# Cellular Senescence and Inflammaging in the Bone: Pathways, Genetics, Anti-Aging Strategies and Interventions

**DOI:** 10.3390/ijms25137411

**Published:** 2024-07-05

**Authors:** Merin Lawrence, Abhishek Goyal, Shelly Pathak, Payal Ganguly

**Affiliations:** 1School of Biological and Chemical Sciences, University of Galway, H91W2TY Galway, Ireland; 2RAS Life Science Solutions, Stresemannallee 61, 60596 Frankfurt, Germany; 3Observational and Pragmatic Research Institute, 5 Coles Lane, Oakington, Cambridge CB24 3BA, UK; 4Leeds Institute of Rheumatic and Musculoskeletal Medicine, University of Leeds, Leeds LS9 7JT, UK

**Keywords:** cellular senescence, bone, aging, inflammaging, seno-therapeutics, anti-aging strategies, bone regeneration, therapeutic interventions

## Abstract

Advancing age is associated with several age-related diseases (ARDs), with musculoskeletal conditions impacting millions of elderly people worldwide. With orthopedic conditions contributing towards considerable number of patients, a deeper understanding of bone aging is the need of the hour. One of the underlying factors of bone aging is cellular senescence and its associated senescence associated secretory phenotype (SASP). SASP comprises of pro-inflammatory markers, cytokines and chemokines that arrest cell growth and development. The accumulation of SASP over several years leads to chronic low-grade inflammation with advancing age, also known as inflammaging. The pathways and molecular mechanisms focused on bone senescence and inflammaging are currently limited but are increasingly being explored. Most of the genes, pathways and mechanisms involved in senescence and inflammaging coincide with those associated with cancer and other ARDs like osteoarthritis (OA). Thus, exploring these pathways using techniques like sequencing, identifying these factors and combatting them with the most suitable approach are crucial for healthy aging and the early detection of ARDs. Several approaches can be used to aid regeneration and reduce senescence in the bone. These may be pharmacological, non-pharmacological and lifestyle interventions. With increasing evidence towards the intricate relationship between aging, senescence, inflammation and ARDs, these approaches may also be used as anti-aging strategies for the aging bone marrow (BM).

## 1. Introduction

Advancing age and age-related changes present major health challenges globally. It has been closely linked with several conditions, age-related diseases (ARDs) and is an established factor that contributes to reduced quality of life (QOL) in the elderly. In fact, as per the information shared by the World Health Organization (WHO), in 2020 the number of people over the age of 60, exceeded the number of children below the age of 5 worldwide [1]. While it must be noted that not everyone over the age of 60 lives with a diseased condition, advancing age has been identified as one of the major contributing factors to several diseases, including cancer [2,3,4], neurodegenerative [5,6,7] and musculoskeletal diseases [8,9,10].

All of these diseases are denoted by physical challenges in the elderly by increased frailty and reduced ability for repair and day-to-day functioning. At the cellular level, the common denominator apart from advancing age, can be pinned down to cellular senescence. Cellular senescence can be defined as an irreversible loss of a cell’s proliferative capacity [11] and often results from or into one of the hallmarks of aging including DNA damage, mitochondrial dysfunction and stem cell exhaustion [12,13]. Specifically in terms of the aging bone, the elderly are faced with reduced mobility, stiffness and pain in joints and susceptibility to diseases like osteoarthritis (OA), osteoporosis (OP) and rheumatoid arthritis (RA), with increased vulnerability towards to fracture impacting millions worldwide [14,15,16,17,18].

The bone by itself is an extremely dynamic organ, with a hard exterior and a tremendously active bone marrow (BM) that houses multiple types of immune cells, two types of progenitor or stem cells, various cytokines and growth factors [19]. Alongside housing these cells, the BM is the seat of stem cell formation that gives rise to the different cell types and provides the right environment for cellular communication necessary for its functioning [20]. However, with advancing age the functionality of bone has been known to deteriorate and cellular senescence within the bone reduces its ability to regenerate and repair [21,22,23]. Briefly, the number of stem cells and progenitor cells decline with advancing age and there is a decline in bone strength and bone density, also known as age-related bone loss [19,24]. The BM becomes more adipogenic while compensating for bone formation and thus has reduced osteogenic capacity [25]. Additionally, there is myeloid skewing wherein an increased number of hematopoietic cells shift toward the myeloid lineage with advancing age. This has also been linked to age-related malignancies [26]. We have previously discussed these in great detail, including the aging BM and its immune compartments in 2021 [27].

Another key concept that underpins these ARDs at the cellular level and has come to light more recently is inflammaging. It is defined as the ‘chronic low grade inflammation that occurs with advancing age’ [28,29,30] and has also been identified as one of the most potent risk factors that contribute to morbidity and several ARDs [31,32,33]. Due to their role in several diseases, both cellular senescence and inflammaging are increasingly being investigated at the genetic level and have been hypothesized as potential targets for anti-aging and regenerative strategies [34]. Some of these include senolytic drugs that target the apoptosis of identified senescent cells and their lysis [35,36], senomorphic drugs that aim to reduce the effect of pathways and factors associated with senescence, without killing these cells [37,38]. Additionally, other biomaterials and rejuvenation procedures include the use of biologics, such as platelet-rich plasma (PRP) [39,40] or methods like parabiosis that involves the surgical union of two living organisms such that it resembles a single physiological system [41,42]. Additionally, a natural step towards a healthy lifestyle that includes a balanced diet and regular exercise has always shown benefits for bone health [43].

In this review, we dissect the recent literature on bone aging and senescence, focusing on the effect of inflammation and inflammaging in association with bone-related ARDs. We then outline the pathways that have been established to be involved in cellular senescence and inflammaging in the bone and related orthopedic diseases like OA, OP and RA followed by their genetic landscape. We then combine this knowledge with therapeutic approaches by outlining different ways of combatting senescence using seno-therapeutics, other potential anti-aging strategies targeting cellular senescence and changes in lifestyle. Finally, we discuss the limitations, challenges and future perspectives of this dynamic field of bone aging and senescence.

## 2. Cellular Senescence and Inflammaging in the Bone

The term cellular senescence was first used by Hayflick et al. in 1961 [11] after which it has increasingly been used and explored in relation to advancing age, cancer and other age-related bone diseases [2,22,44,45]. Biologically, cellular senescence is an essential phenomenon for processes like tumor suppression, wound healing and tissue fibrosis; however, increasing evidence indicates the harmful effects of accumulated senescence and its characteristic senescence associated secretory phenotype (SASP) [46]. Characteristic features of senescent cells include enlarged cells, reduced proliferation and differentiation abilities, telomere shortening, chromatin remodeling, increase in cell cycle regulators like p16, p21 and p53 and an increase in senescence-associated beta-galactosidase (SA-β-gal) and SASP. SASP includes several proteins, pro-inflammatory cytokines and chemokines like interleukin 6 (IL6), interleukin 8 (IL8), interleukin 1 beta (IL1β), tumor necrosis factor alpha (TNFα), C-C motif chemokine ligands 2, 3, 5, 8 and many others. Overall, these senescence and SASP markers across different tissues have recently been further outlined and categorized thoroughly by Suryadevara and colleagues [47]. These have been shown to negatively impact the proliferation, growth and functional capacity of cells [48].

Cellular senescence may originate as a result of physiological insults, oxidative stress due to reactive oxygen species (ROS), radiation and nutrient deprivation. All of these factors can cause DNA damage that may lead to cell cycle arrest, rendering a cell to become senescent. These senescent cells then release SASP, which creates a microenvironment unsuitable for cell proliferation, growth, survival and eventually leads to cell damage and death. This has been indicated below in Figure 1. Interestingly, SASP has also been associated with being the fuel for chronic and systemic low-grade inflammation during aging–now known as ‘inflammaging’ [31]; both of which contribute to ARDs and related mortality.

With respect to senescence in the bone, several in vitro and ex vivo studies have demonstrated increased senescence and age-related decline in the number and function within the stem cell compartments in the BM [24,49,50]. Radiation-induced senescent osteocytes were found to degenerate the differentiation potential of BM MSCs in vivo via the paracrine pathway [51]. In a doxorubicin induced senescence model, hypoxia inducible factor (HIF)-2α and p21 were found to be elevated and the osteoblast differentiation was inhibited. The same study along with in vivo results found that HIF-2α acts as an intrinsic factor for age-related bone loss [52].

Similarly, several articles have attempted to connect the links between aging, senescence and inflammaging, considering inflammaging is a relatively new concept [21,31,32]. However, to date, experimental evidence that is solely focused on bone senescence and inflammaging remains limited. Interestingly, reports of both cellular senescence and inflammaging within the bone have been addressed more frequently in conditions like OA [53,54], OP [55,56] and sometimes RA [57,58]. In the following sections, we discuss the pathways, genetics and anti-aging strategies taking inspiration from evidence presented in these age-related orthopedic conditions including OA, OP and RA.

## 3. Pathways Involved in Senescence and Inflammaging in the Bone

Numerous pathways and cycles are involved in the normal functioning of a body, which undergoes age-related changes that impact different tissues. Specifically, for pathways involved in senescence across tissues, Wang et al. have recently discussed them in a recent review article [59]. For this review, we will focus on the pathways that have demonstrated their involvement in cellular senescence, specifically in the bone and associated ARDs. In particular, cellular senescence has been observed and reported in bone progenitor cells known as mesenchymal stem cells (MSCs) [60], osteoblasts [56] and differentiated osteoblasts known as osteocytes [61]. These largely include pathways involved in cell cycle, growth promotion and DNA damage due to oxidative stress via the STING pathway. MSCs, osteoblasts and osteocytes have all been indicated to potentially be impacted by the pathways outlined below.

### 3.1. Cell Cycle Arrest: p16, p21, p53

p16, p21 and p53 are key markers involved in cell cycle and DNA damage response (DDR), acting as cell cycle checkpoints [62]. Together, these three markers are the most commonly compromised ones in conditions like cancer, making them essential for programmed senescence, genomic stability and repair responses [62]. Senescent cells have been shown to have upregulated levels of cyclin-dependent kinase inhibitors (CKIs) like p16 and p21, indicated by upregulated mRNA levels of p16^INK4a^ and p21 in elderly humans compared to young women [63]. p53 followed by p16, has been indicated to play a stronger role in senescence response to telomere dysfunction than p21 [64]. Interestingly, Chandra et al. found that targeting p21 but not p16-positive senescent BM cells in vivo prevented OP in mice [65]. BM MSCs from myelodysplastic syndrome (MDS) were found to demonstrate senescence phenotypes (enlarged cells, reduced proliferation, increased SA-β-gal) via activation of p21/p53. Interestingly, they too did not find any changes in the levels of p16 or pRb in MSCs from MDS patients [66]. Similarly, MSCs from patients with systemic lupus erythematosus (SLE) were also found to be senescent via p21/p53 activation [67]. The overall role of p16, p53 and p21 in cell cycle in relation to bone function is indicated in Figure 2.

#### 3.1.1. p16-Rb Pathway

p16, also known as INK4a, is a protein encoded by the gene cyclin-dependent kinase inhibitor 2A (CDK2NA), which is an essential inhibitor of cyclin-dependent kinase activity and a tumor suppressor gene. It has several functions in different stages of the cell cycle and is necessary for regulating uncontrolled cell proliferation in majority of the cell types [68]. It is essential for the cell cycle’s transition from G1 toS phase. In particular, p16 binds to CDK4/6 and prevents the phosphorylation of the Rb protein. It stops the formation of the cyclin D–CDK4/6 complex. In turn, hypo-phosphorylated Rb binds to E2F transcription factors, preventing cell cycle progression, thus leading to G1 cell cycle arrest, further preventing the transition into the S phase [69].

OA is a well-acknowledged age-related degenerative disease where senescence plays a significant role in its progression [53,54,70,71]. Philipot et al. demonstrated the accumulation of p16^INK4a^ positive cells in an in vitro experiment using mature chondrocytes for the progression of OA [72]. In response to pro-inflammatory cytokines like IL6, IL8, and IL1β, chondrocytes produce matrix-remodeling regulatory matrix metalloproteases (MMPs) like MMP1 and MMP13, exhibiting the SASP characteristics. Together with these SASP factors, multiple pathways lead to the activation of p16, which in turn pushes them towards senescence. In vitro studies by Farr et al. indicated that growing osteoblasts in media conditioned with cellular senescence, impaired osteoblast mineralization, leading to increased osteoclast genesis. The same study with in vivo experiment in mice showed that the clearance of p16^INK4a^ positive cells attenuated age-related bone loss [73].

#### 3.1.2. p53-p21 Pathway

p21, also known as CIP1 and WAF1 is a protein encoded by the gene *CDKN1A*. It can induce cellular senescence in p53 dependent and p53 independent pathways. *p53* is a tumor suppressor gene, which has the main implication in the protection of the DNA integrity of the cell [74]. In p53-dependent pathway, ATM-Chk2 or ATRChk1 pathways activate p53, thereby upregulating its downstream target p21 expression [75], where p21 binds to CDK-2, 4 and 6 and inhibits the cyclin–CDK complex formation, resulting in RB–E2F complex formation,. Consequently, this prevents cell cycle progression by inducing G1arrest and G2/M [74,76,77]. In the p53-independent pathway, p21 independent of p53 inhibits cyclin-CDK complex which further prevents cell cycle progression via Rb-E2F complex.

Additionally, p21 binds to and inhibits the activity of proliferating cell nuclear antigen (PCNA), a subunit of DNA polymerase, which results in the termination of DNA replication [68,78]. Englund et al. demonstrated the role of p21 in inducing cellular senescence via DDR and inflammatory signaling pathways in transgenic mice, with p21 overexpression in skeletal muscles. p21 overexpression in senescent cells corroborated with tissue fibrosis, low levels of skeletal muscle mass and reductions in physical function [79]. Loss of p53 in mice BM MSCs has been reported to alter bone remodeling and potentially impact cancer related bone modeling via negative regulation of osteoprotegerin (OPG) [80].

### 3.2. Growth Promotion Pathways—mTOR and, SIRT-1

#### 3.2.1. mTOR

mTOR (mammalian target of rapamycin) is a critical negative regulator of autophagy. It is activated downstream of PI3K kinase and Akt kinase to inhibit autophagy. It forms two catalytic subunit protein complexes following a number of steps: mTORC1, which regulates cell growth and metabolism and mTORC2, which controls proliferation and survival [81]. mTORC1 actively inhibits autophagy by phosphorylating ULK1, thereby preventing its activation by AMPK (Figure 3). Senescent cells show failure in autophagy due to mitochondrial dysfunction, another hallmark of senescent cells [82].

Zhang et al. showed a correlation between the overexpression of mTOR and disease progression using an in vivo OA mouse model [83]. Its ablation at the genetic level alleviated the OA pathogenesis via the upregulation of autophagy [84,85]. Pan et al. showed the attenuation of cartilage degradation in the OA mice model by rectifying the autophagy inhibition of chondrocytes via PI3K/AKT/mTOR signaling [86]. Cheng et al. demonstrated inhibition of autophagy improved OA-related degeneration in vivo in femoral condyles of rabbits and in vitro in human chondrosarcoma cells [84].

#### 3.2.2. SIRTs

SIRTs (Sirtuins) are a family of nicotinamide adenine dinucleotide (NAD+)-dependent protein deacetylases, which regulate a comprehensive range of cellular processes ranging from cell metabolism, development and cellular senescence [87,88]. It comprises of seven members from SIRT 1–SIRT 7.

Among them, SIRT-1 is critical in maintaining cartilage health by promoting chondrocyte survival and ECM stabilization. It has been connected to longevity and extended lifespan [89]. Additionally, SIRT-1 activation has demonstrated a protective effect in OA by enhancing the trabecular and subchondral bone and by inhibiting chondrocyte apoptosis [90]. SIRT-1 is involved in multiple pathways, such as NF-κB, AMPK, mTOR, p53, PGC1α and FoxOs have been implicated in inflammation, mitochondria biogenesis, autophagy, energy metabolism, oxidative stress and cellular senescence [88]. In senescent cells, SIRT-1 is identified as a nuclear autophagy substrate degraded by being transported to cytoplasmic autophagosomes through LC3 recognition during senescence [91]. Batshon et al. showed cleavage of SIRT-1 into N-Terminal (NT) and C-Terminal (CT) by cathepsin B. Increased serum NT/CT SIRT-1 ratio reflected the early stage of OA and cellular senescence in chondrocytes [92]. Zhou et al. found that SIRT-1 regulated osteoblast senescence upon exposure to cadmium. Interestingly, the exposure simultaneously decreased SIRT-1, increased osteoblast senescence as well as increased p53, p16 and p21, which triggered DDR [93].

### 3.3. Reactive Oxygen Species (ROS)-Induced DNA Damage (Type 1 IFN and the STING Pathway)

During aging, chronic low-grade systemic inflammation produces increased ROS levels, which lead to oxidative stress [94]. Oxidative stress induces DNA damage found at higher levels in aged cells and has been associated with diseases in almost every organ [95,96]. The release of mitochondrial or nuclear DNA from these damaged cells activates the innate immune system. One part of the innate immune activity is the cyclic GMP–AMP synthase (cGAS) stimulator of interferon genes (STING) signaling pathway. cGAS enzyme triggered by the detection of leaked DNA further catalyzes the production of cyclic GMP-AMP (cGAMP). Subsequently, cGAMP interacts with STING, activating the STING pathway via its various downstream targets, such as TANK-binding kinase 1 (TBK1), interferon regulatory factor 3 (IRF-3), and IkappaB kinase (IKK). Consequently, the induction of type 1 interferons commences. Additionally, the STING pathway regulates nuclear factor-kappa-light chain enhancer of B cells (NF-κB) signaling in chondrocytes implicated in SASP and senescence. DDR-induced type 1 interferon has been indicated to promote senescence and inhibit stem cell function from both hematopoietic and stromal compartments in the BM [97,98]. The cGAS–cGAMP-STING axis has now been suggested as the missing link between DNA damage, inflammation, cellular senescence and cancer [99]. Figure 4 below demonstrates this pathway in an OA joint.

STING pathway induces ECM degradation via upregulation of matrix-degrading enzymes (ADAMTS5 and MMP13) as a part of SASP [100]. STING pathway activation is associated with cellular senescence and apoptosis [101]. Guo et al. demonstrated that the knockdown of STING using lentivirus in an in vivo OA mouse model improved senescence, apoptosis and ECM imbalance in chondrocytes [100]. Owing to their prominent role in inflammation and SASP factors elevation, they are becoming an attractive target for treating cellular senescence.

## 4. Genetics of Cellular Senescence in the Bone

During cell development, cellular impairments such as DNA damage, telomere shortening or dysfunction, oncogene activation or loss of tumor suppressor capabilities, epigenetic modifications and organelle destruction may occur. All the aforementioned processes contribute to cellular senescence, but in particular towards DDR [102]. Given that the bone tissues produce 95% of the body’s cells, numerous studies have utilized bone biopsies to investigate which, if any, genes play a crucial role in cellular senescence and what effect this has in the context of pathophysiology [103].

SASP, as described in Section 2 above comprises of a distinct secretory pattern, causing cell cycle termination. The diverse and tissue specific SASP factors typically consist of adhesion molecules, chemokines, cytokines, growth factors and lipid components [102]. These factors can have both local and systemic effects that can result in a variety of ARDs. In view of the molecular mechanisms and underlying pathways outlined above, the question then arises as to which genetic aberrations or abnormalities are contributive or causative of such phenomena.

### 4.1. Key Genes Associated with SASP

SASP research has indicated that age-related inflammatory responses, wound healing and cancer progression can all be mediated by the secretion of pro-inflammatory agents, growth factors, chemokines and proteases by aged cells [104,105]. Several conventional SASP proteins have been observed to be increased in aged bone matrix extracellular vehicles, including TGF-β2, OPG, MMP9, tissue inhibitor of metalloproteinase 1 (TIMP1), macrophage migration inhibitory factor 1 (MIF1), peroxiredoxins (PRDXs) and insulin like growth factor binding protein 3 (IGFBP3). Among these, TGF-β2 and OPG play a key role as mediators of bone metabolism; serum levels of both have been found to positively correlate with indicators of bone turnover [106]. Osteocytes and osteoblasts both express TIMP1, a tissue inhibitor of MMPs that controls MMP levels to maintain the equilibrium of bone matrix breakdown [107]. Additionally, TIMP/MMPs are able to modify the extracellular matrix to accommodate the immunological influx [108]. MIF1 is known to activate a number of transcription factors and stress kinases, such as adenosine monophosphate or AMP- activated protein kinase (AMPK) and B NF-κB, which can mediate inflammatory and tumorigenic signaling [109]. The way that aged bone cells influence the environment of nearby and distant cells and systems may therefore be explained by the elevated production of these SASP proteins [110].

Seven particular gene sets have been published specifically associated with senescence, the most comprehensive one being SenMayo, surpassed the others in identifying senescent cells with aging across tissues and species and in exhibiting responses to senescent cell clearance (based on normalised enrichment scores (NES) and *p* values) [102].

### 4.2. NGS Studies on Bone Aging

We have previously described key NGS studies targeting the aging BM [27]. Specifically for studies that investigated parameters for bone senescence, we list five key papers in Table 1 below.

A review of the literature pertaining to the sequencing of genes within the BM d-rived cells exhibits that the most impactful study in view of bone senescence was carried out by Saul et al. in 2022 [102]. This study took into account the lack of a defined ‘senescence gene set’ within the field, thus creating the 125 gene panel called ’SenMayo’, consisting of previously discovered senescence and SASP-associated genes. The authors aimed to identify commonly regulated genes in different age-related datasets (n = 15), using a transcriptome-wide approach that included whole-transcriptome as well as single cell RNA sequencing (scRNA-seq). In order to determine if senescence and SASP associated pathways were enriched with human aging and utilizing whole-bone biopsies, Saul et al. identified in particular, that genes regulating inflammatory mediators, such as *NFKB1*, *RELA*, and *STAT3*, were enriched. Additionally, *CDKN1A/p21Cip1*, *CCL2*, and *IL6* were upregulated in both elderly cohorts, as the authors initially predicted.

### 4.3. Potential Pathways to Target in Bone Aging

Identification of upregulated and downregulated genes within bone senescence leads to the identification of key pathways within this phenomenon. Whilst it is challenging to eliminate pathways solely based on genetic influences, key molecular targets can be highlighted and potentially targeted in bone aging.

Saul et al.’s genetic panel consisted of several SASP factors (n = 83); transmembrane (n = 20) and intracellular (n = 22) proteins [102]. The key regulatory elements of the SenMayo genes, i.e., *NFKB1* motif and *BCL3*, the latter of which is a key transcriptional coactivator for NF-κB, represented the leading transcription factor for a majority of SASP genes. This strongly suggested that targeting NF-κB signaling, a master regulator of gene transcription, is one of the cell’s most efficient ways to influence important and varied biological processes, including senescence [115].

This aging-related upregulation of inflammatory proteins (i.e., inflammaging), has been suggested to be the root cause of SASP, which releases inflammatory factors into the environment and causes neighboring cells to undergo bystander senescence or potentially change into pre-cancerous cells. The SASP phenotype activation occurs in conjunction with the overexpression of inflammatory markers, which is also present in senescence but is not related to terminal replication. The cause of this is unknown. That inflammation, which can cause tumorigenesis, is elevated during senescence—a process that suppresses tumor growth—seems incongruous. This could be an illustration of antagonistic pleiotropy; senescence prevents the development of cancer in its early stages but, as the organism ages, the secretory phenotype associated with senescence becomes detrimental [112]. Thus, it is becoming increasingly important to have regeneration and anti-aging strategies to combat cellular senescence in the aging bone.

## 5. Combatting Senescence in the Bone

Approaches for regeneration of senescent cells in the bone may be classified as three types (Figure 5); 1. pharmacological with seno-therapeutics; 2. non-pharmacological with biologics and biomaterials; and 3. based on lifestyle. Pharmacological approaches use various types of drugs or conditions that reduce or eliminate senescence [37,116]. Non-pharmacological approaches use biomaterials like shell nacre and biologics like platelet-rich plasma (PRP) that have demonstrated anti-aging and anti-senescent properties in aged cells and in ARDs [117,118]. Finally, the lifestyle approach includes changes in diet, such as intermittent fasting or a diet high in antioxidants, to combat ROS-mediated senescence and regular exercises to naturally ameliorate cellular senescence [119,120].

### 5.1. Pharmacological Approaches

Seno-therapeutics are pharmacological interventions targeting senescent cells or SASP. It consists of three classes of therapeutics namely senolytics, senomorphics that are more established and seno-inflammation blockers that are currently being tested in vitro.

#### 5.1.1. Senolytics

This is the class of therapeutics that may potentially eliminate senescent cells to prevent or alleviate several age-related conditions such as senile OP [121]. Research indicates that the selective elimination of senescent cells delays disease progression in OA and other ARDs [122]. One of the earliest studies published by Zhu et al. analyzed the differential gene expression in senescent versus non-senescent human pre-adipocytes and noticed the upregulation of genes involved in senescent cell anti-apoptotic pathways [123]. The authors identified over 46 potentially senolytic compounds that silenced key pro-survival anti-apoptotic genes such as *EFNB1/3*, *PI3Kδ*, *p21* and *BCL-xL* (individually and in combination). The most effective compounds were Dasatinib (SRC/tyrosine kinase inhibitor) and Quercetin (natural flavonoids that interact with BCL-2 and P13K isoforms). Dasatinib, an FDA-approved cancer drug proved to be effective against senescent human preadipocytes and quercetin was effective against senescent human endothelial cells. They also showed that the combined therapy of Dasatinib and Quercetin proved more effective in eliminating senescent mice mesenchymal embryonic fibroblasts [123]. Since then, several compounds have been screened and tested in clinical trials as summarised by Chaib et al. [116].

Several classes of senolytics have been reported since then and have significantly shown to increase lifespan and mitigate ARDs [124]. Highly targeted treatments such as ABT-263, also known as the chemotherapeutic drug Navitoclax, is a BCL-2 pathway inhibitor that has been shown to cause thrombocytopenia and neutropenia under small doses [125]. Interestingly, while Chang and colleagues demonstrated rejuvenated BM HSCs in vivo by using ABT-263 to clear senescent cells [126], another study, which evaluated the effect of navitoclax on mice, showed trabecular bone loss and musculoskeletal dysfunction [127]. These studies thus warrant a closer look at the strategies adopted for targeting different components of the senescent cells anti-apoptotic pathways (SCAPs). Higher doses targeting single points in a pathway might have more off-target, detrimental effects and therefore a combination of different inhibitors in low-dosage can prove to be safer and more efficacious [128].

#### 5.1.2. Senomorphics

These aim to target the pathways and factors associated with senescence without killing the senescent cells. Also known as SASP inhibitors, these agents try to make senescent cells mimic young cells by modulating their signal transduction mechanisms. The most popular senomorphics are rapamycin, resveratrol, NF-κB, JAK/STAT and p38MAPK inhibitors [37]. Rapamycin, also known as sirolimus, is an inhibitor of pro-senescent mTOR signaling. It acts by reducing the phosphorylation of S6K and 4E-BP downstream of TORC1 (discussed in Section 3.2.1). TORC1 forms a multiprotein complex with TORC2 and is regulated by mTOR [129]. Although effective in in vivo studies, side effects of rapamycin such as nuclear factor erythroid 2-related factor 2 (Nrf2) pathway activation and NF-κB suppression have been reported. Even though off-target, these effects supplement the senomorphic effect of rapamycin [130]. Resveratrol is a plant antitoxin and antioxidant that activates the SIRT-1 pathway, discussed previously in Section 3.2.2. SIRT proteins are key regulators of transcriptional processes and pathways involved in senescence and aging. However, with increasing age, their expression and functional activity decreases [91].

Studies show that regulated doses of resveratrol can suppress SASPs by activating P13K-aKT signaling in endothelial progenitor cells and BM stromal cells [131,132]. However, higher concentrations can induce senescence and apoptotic death. This has led to researchers investigating other novel SIRT activators such as SRT1720, STAC-5/9/10, SCIC2 and SCIC2.1 [133,134,135]. These anti-aging compounds are more stable and have demonstrated better bioavailability. However, it should be noted that the diet of the organisms is vital to its efficacy. Resveratrol performs better when supplemented with a high-fat diet, whereas the second-generation SIRT activators have high efficacy with a standard diet.

Other senolytic compounds currently being investigated are p53 activators (UBX0101) and PPARα agonist fenofibrate [136,137]. Another attractive target is p38MAPK. Mitogen-activated protein kinase (MAPK) modules regulate cell proliferation, differentiation and apoptosis processes. p38MAPK is a sub-family of this pathway and is activated in response to stress. Its ability to trigger cell-cycle arrest and induce senescence has been harnessed to create inhibitor compounds such as SB203580, UR-13756 and BIRB-796. They have been shown to block SASP secretion in senescent cells. SB2023580 suppresses p16 signaling which mediates cell cycle arrest and SA-β-gal activity [20]. R-13756 and BIRB-796 are second-generation p38 inhibitors with higher specificity as shown in vivo using human fibroblasts [138]. Interestingly, p38 MAPK inhibitors (MK2i) have indicated the potential to have anti-inflammatory effects in RA, which is an autoimmune and degenerative disease of joints. However, further studies are needed to confirm the same [139,140]. Another popular anti-aging compound is JAK/STAT inhibitor ruxolitinib, an FDA-approved drug used in myeloproliferative diseases. It has been shown to improve age-related bone loss in mice attributed to its ability to inhibit SASP secretions [73]. Nevertheless, with the current black box warning against JAK inhibition, alternative approaches for targeting senescence are needed [141].

#### 5.1.3. Seno-Inflammation

This term refers to a dysregulated immune response that leads to chronic inflammation. It was proposed by Chung et al. in 2019 to describe the steady-state age-related senescent inflammation that exacerbates chronic diseases, associating with the concept of inflammaging [142]. On a molecular level NF-κB, signaling plays a major role in inflammation. It upregulates chemokines, interleukins (IL2, IL6 and IL-1β), adhesion molecules and c-reactive protein (CRP) triggering immune system activation [143]. On a cellular level, the dysregulated macrophage activity has been associated with aging in humans. Toll-like receptor (TLR) signaling, interferon-gamma (IFN)-γ activity, and NF-κB signaling changes are downstream markers of macrophage activity in senescent cells. Proteoglycan-4 (PRG-4/lubricin) is a modulator of TLR signaling and can regulate inflammatory response in vitro and in in vivo rat models of OA [144]. PRG-4 binds to and suppresses TLR2 and TLR4 activity in human OA synovial fluid, making it a potential anti-inflammatory therapeutic target [145]. Senescence-associated secretomes and growth factors are also being explored as therapeutics for reducing seno-inflammation. Platas et al. used a conditioned medium from MSC secretomes and cultured OA chondrocytes. The cells reduced inflammatory stress, assessed by markers IL-1β and SA-β-gal [146]. Growth hormones such as pegvisomant inhibit growth hormone/insulin-like growth factor-1 (IGF1) axis and modulate inflammation, making it an attractive therapeutic agent [147].

A few examples of pharmacological approaches with senolytic and senomorphic compounds that have been investigated are outlined in Table 2 below.

### 5.2. Non-Pharmacological Approaches

#### 5.2.1. Materials for Bone Regeneration via Senescence Reduction

Regenerative medicine plays a crucial role in promoting cell neogenesis in the context of reducing senescence and promoting bone health. While seno-therapeutics have shown to increase the regenerative capacity of the bone; tissue engineering scaffolds, bioactive and biocompatible biomaterials can also provide the necessary microenvironment for tissue regeneration, growth and repair. They do so by improving the drug delivery and the temporary effect of the drug in vivo [117,148,149,150]. Hydrogels have been tested to provide an effective microenvironment for tissue growth and cell differentiation [151,152,153]. Quercetin-loaded hydrogels caused clearance of senescent cells in aged rats for a prolonged period thereby improving the in-vivo lifespan and delivery of the drug [154]. Microspheres loaded with drugs or growth factors serve as carriers for time-dependent and location-specific release. Thus, these platforms provide a major improvement in ameliorating side effects from seno-therapeutic drugs [155]. He et al. tested the efficacy of PEGylated (amalgamated with polyethylene glycol) hydrogels loaded with rapamycin on senescent MSCs. The oxidative function of this system was able to delay senescence by scavenging intracellular ROS [153]. New investigation on novel materials like shell nacre 3D scaffolds have demonstrated enhanced proliferation of MSCs from older donors that was comparable to MSCs from younger donors, in spite of previously observing higher number of SA-β-gal cells in the older MSCs in 2D cell culture [117].

Another exciting anti-aging compound suitable for bone regeneration is collagen. Collagen is the structural protein in the ECMand is biocompatible, water-soluble and easily metabolized. It can be extracted from abundant marine organisms, making it a suitable biomaterial for bone regeneration [156]. Collagen has been shown to increase mineral bone density and high osteoblastic activity, promoting self-renewal abilityand enhancing osteogenic capacity of the bone in vitro using rat BM MSCs and osteoblastic cells [157,158]. It is known that there is a high accumulation of ROS and associated inflammation in aging models. In 2019, Zhou et al. functionalized a titanium mesh-based porous scaffold using polypyrrole. Polypyrrole is an electroactive conductive polymer that can induce physical and chemical changes in response to electrical signals. They used this conductive polymer to scavenge ROS by providing free electrons. To increase the osteoconductivity of the biomaterial, they added hydroxyapatite nanoparticles to the scaffold. To increase the hydrophilicity and drug-loading efficacy of polypyrrole, they supplemented this system with polydopamine nanoparticles. Together when the polypyrrole polydopamine hydroxyapatite film was coated on a titanium mesh-based scaffold and tested on RAW264.7 macrophages, the scavenging activity of ROS was significantly reduced [159].

#### 5.2.2. Anti-Aging Strategies Using Biologics

The use of biological adjuvants to study bone healing has yielded several treatments that induce signaling and anti-inflammatory effects. Stem cell therapy using MSC extracted from the BM has proven to deliver significant clinical advantages in tissue regeneration. MSCs secrete proteomes that regulate the immune system and demonstrate anti-apoptosis, anti-oxidation, cell homing and differentiation abilities [36]. This has been used to activate repair mechanisms in cartilage, muscle and bone and thus their potential to treat ARDs is highly probable [160,161]. MSCs have been an area of great interest within the scientific as well as the medical community for use in bone regeneration [162,163,164,165,166].

Blood and blood-related factors are also contributing towards reversing age-related molecular and cellular changes in blood, muscle, bone and nervous system. Parabiosis in the context of aging is the technique of surgically joining the circulatory system of a young organism with an aged organism to study the reverse effects of aging [167]. Heterochronic parabiosis (HP) performed on mice showed that youthful circulation increased the capacity of bone healing in aged mice with tibial fractures. This was marked by the increase in type 1 collagen production expressed in osteoblasts and modulation of the β-catenin pathway [168]. Thus, HP offers a unique but promising paradigm to tackle ARDs.

PRP and other platelet-derived biologics are currently gaining popularity as effective treatments for OA, fractures and enhancing wound healing. Due to its ease of harvesting and preparation, PRP treatments have been touted as an accessible form of treatment harboring anti-inflammatory effects [39]. It has often been used for enhancing cellular functionalities in OA [164,169,170,171] and has also been compared with collagen or hyaluronic acid in the treatment for degenerative diseases like OA with similar or better results [171,172,173]. PRP has been reported to reduce SASP factors like IL6, IL1β, TNFα [173] along with reducing inflammation in inflamed joints potentially targeting inflammaging [174]. Especially with respect to OA, PRP has been observed to reduce inflammation, alleviate pain and reduce pro-inflammatory cytokines in patients [175,176]. However, the lack of consistent results has deterred the progress of PRPs as clinical interventions. A systemic review done in 2023 on the comparison of PRP treatment versus approved treatments for knee OA stated that the current data from trials were inconsistent, biased and hence of low quality [177]. This warrants better methodological approach in the execution, reporting, and analysis of clinical trials, along with the need for standardized protocols for the application of PRPs.

A few examples of non-pharmacological approaches with biomaterials and biologics investigated have been outlined in Table 2 below.

### 5.3. Lifestyle Approaches as Regeneration and Anti-Aging Strategies

Healthy diet approaches along with exercise have been a sought after approach for managing several health conditions and contribute towards healthy aging [119,178]. The concepts of calorie restriction, intermittent fasting, fiber and protein rich diet along with exercises is possibly the most long standing and effective way to ensure that the negative effects of senescence are reduced in the elderly and encourages healthy aging. Consumption of naturally occurring compounds including anti-oxidants, polyunsaturated fatty acids (PUFA), minerals, vitamins and essential amino acids are key to longevity [179]. The combination of a healthy diet with regular exercises like walking and strength training in ‘blue zones’ has shown us that people can live up to a century and more [180].

A recent study investigated the effect of intermittent fasting for 30 days on n = 25 young healthy males by examining their mRNA levels from the blood samples of these donors. They found that at later time points (a week after 30 days), the level of inflammatory cytokines was reduced, there was induced autophagy and a reduction in the expression of senescent markers discussed in Section 3.1 of this article [181]. Fielding et al. studied the blood from n = 1377 older subjects (aged 70–89 years old) and found that lower functionalities like grip strength, gait, walking etcetera were all associated with high SASP and related markers. They were successfully able to link the SASP phenotype with physical activity in sedentary elderly population [120]. Another study by Maria Fastame indicated that a nutritious diet, strong physical health and well-being along with a sense of community was essential to the Sardinian ‘blue zone’ with several reported centenarians [182].

Martel et al. re-iterate the importance of diet and lifestyle changes and remind us that nutritious food, exercises and adequate sleep can produce anti-senescent effects [119]. They also outline that these findings are not surprising but have not been studied as thoroughly with respect to molecular mechanisms and pathways involved. Thus it is no surprise that an isolated life with excess alcohol consumption and smoking can offset the beneficial effects of any of the anti-senescent strategies [119]. Similarly, sleep deprivation for one night was associated with DDR and activated SASP in n = 29 older adults aged between 61-86 years old [183]. In vivo studies in mice revealed prevention of osteoblast senescence in older female mice when fed a blueberry based diet in their early days [184]. A few of the lifestyle approaches with nutrition and exercise routines investigated have been outlined in Table 2 below.

**Table 2 ijms-25-07411-t002:** Examples of strategies for combatting bone senescence.

Approach	Strategy	Model	Target	Reference
Pharmacological, senolytic	Dasatinib (D) and/or quercetin (Q)	in vitro and in vivo	D- senescent fat progenitors, Q- human endothelial cells and mouse BM MSCs, D + Q—mouse embryonic fibroblasts	[123]
Pharmacological, senolytic	Dasatinib + Quercetin	in vitro and in vivo	Trabecular and cortical bone in mice	[73]
Pharmacological, senolytic	ABT263	In vivo	Senescent bone marrow hematopoietic stem cells in mice	[126]
Pharmacological, senolytic/senomorphic	Zeldronic acid	In vitro and in vivo	Bone	[185]
Pharmacological, senomorphic	Ruxolitinib	In vivo	metabolism	[186]
Non-pharmacological, biomaterial	Shell nacre	In vitro	BM MSCs	[117]
Non-pharmacological, biomaterial	PEGylated hydrogel with Rapamycin nanomicelles	In vitro and in vivo	BM MSCs	[153]
Non-pharmacological, biologics	HA v/s PRP in RCT	In vivo	Knee OA	[173]
Non-pharmacological, biologics	Collagen	In vitro	BM MSCs	[157]
Non-pharmacological, biologics	PRP	In vitro and in vivo	Injured tendons	[174]
Lifestyle	Intermittent fasting	In vivo, n = 25 young males	Senescence markers linked with diet and lifestyle	[181]
Lifestyle	Physical functioning	In vivo, n = 1377 older adults	Senescence biomarkers linked with physical functioning in elderly	[120]
Lifestyle	Diet + exercise	In vivo, n = 12 young males	Senescent cells in skeletal muscle linked with diet and exercise	[187]
Lifestyle	Diet + exercise + community	Mixed methods, n = 57 older adults	Nutritional habits and active lifestyle linked with longevity	[182]
Lifestyle	Sleep deprivation	In vivo, n = 29 older adults	Sleep deprivation linked with senescent markers	[183]

## 6. Conclusions, Limitations and Future Directions

While cellular senescence, ARDs and inflammaging are increasingly contributing to the aging bone and BM, several interventions and anti-aging strategies may be used to combat this effect and aid the regeneration of the bone. We are mindful as authors that aging is a natural progression and that fit and healthy elderly people exist. With this review article, we hope to bring to light that the natural progression of advancing age does not necessarily have to be with diseased conditions. Understanding the molecular mechanisms, genetics and pathways is essential to our knowledge of bone aging. Anti-aging strategies exist and must be optimised for the needs of the individuals.

It is worth outlining that in our quest to discover the connections between bone senescence and ARDs, we found the majority of research evidence in literature in relation to OA, OP and RA. These three ARDs have been reported to be impacted by all of the pathways discussed in Section 3 above. They have also been evidenced with cellular senescence, (progenitor and bone cells for OA and OP as outlined above and T-cells for RA [188]), associated with inflammation and inflammaging and often require surgeries and life-long medication for treatment. This indicated a potentially common root cause for all of these three ARDs, that may be developed into biomarkers and possibly therapeutic targets with further research.

Considering pharmacological approaches for anti-aging strategies, variability and low efficacy of some agents have been observed which can be attributed to the fact that senescent cells have different transcriptional signatures based on their spatial and temporal status. Further screening (chemical, pharmacological and transcriptional) will be warranted to study the precise effects of these molecules in a context-dependent manner. In terms of lifestyle approaches, the aforementioned study was limited by investigating the effect of intermittent fasting only in males, thus their results are not applicable to females and similar studies focused on women are needed [181].

Future investigations focused on human samples from healthy donors across young and older adults, as well as those suffering from ARDs are critical. Additionally, age and gender matched studies of those with lifestyle disorders with a potential to develop ARDs will be of extreme significance. Genetic mapping of these samples and dissecting their pathways will help in early detection of these diseases. This is in turn will be useful in predicting the therapeutic approach (pharmacological or non-pharmacological) best suited for an individual impacted by an ARD. Nevertheless, time and again the benefits of healthy lifestyle have been shown to us and an increasing body of evidence indicates the anti-senescent effect of the same. Thus, irrespective of the presence or absence of disease, nutritious food and healthy lifestyle must be encouraged across age, gender and cultures to combat senescence in the most cost-effective manner.

## Figures and Tables

**Figure 1 ijms-25-07411-f001:**
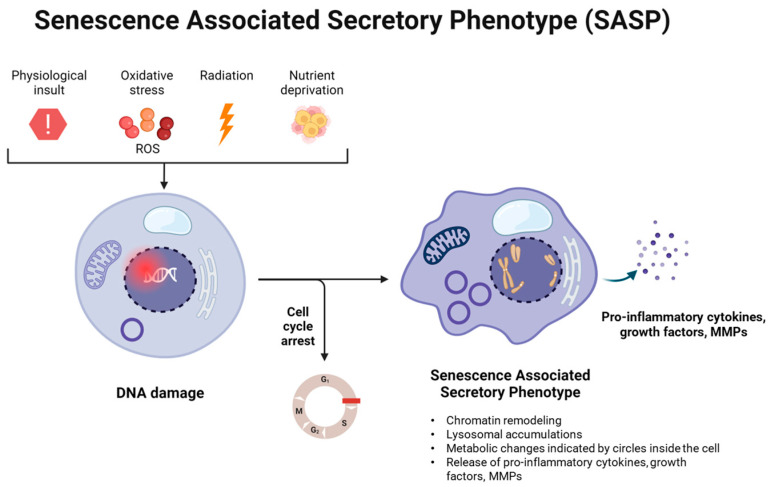
Mechanism of senescence-associated secretory phenotype (SASP) formation.

**Figure 2 ijms-25-07411-f002:**
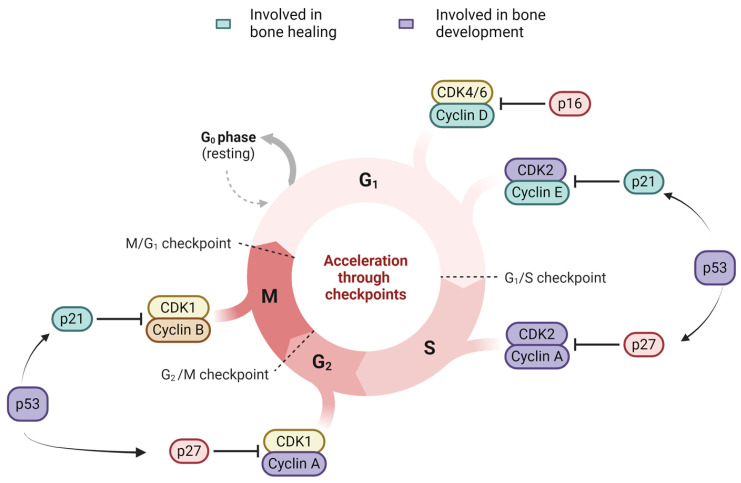
The role of p16, p53 and p21 in bone development and healing in cell cycle.

**Figure 3 ijms-25-07411-f003:**
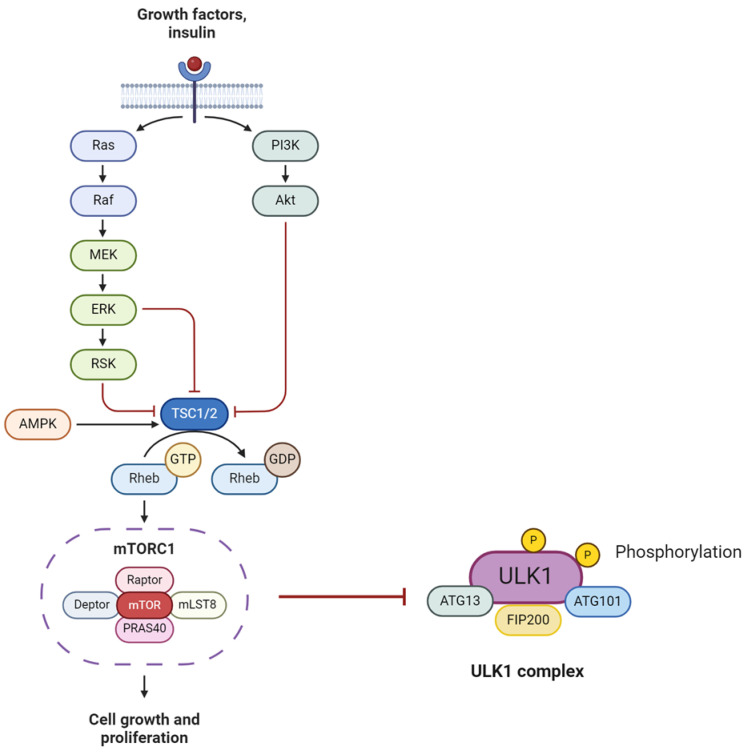
mTOR pathway and formation of subunit mTORC1 and inhibition of autophagy via ULK1 phosphorylation.

**Figure 4 ijms-25-07411-f004:**
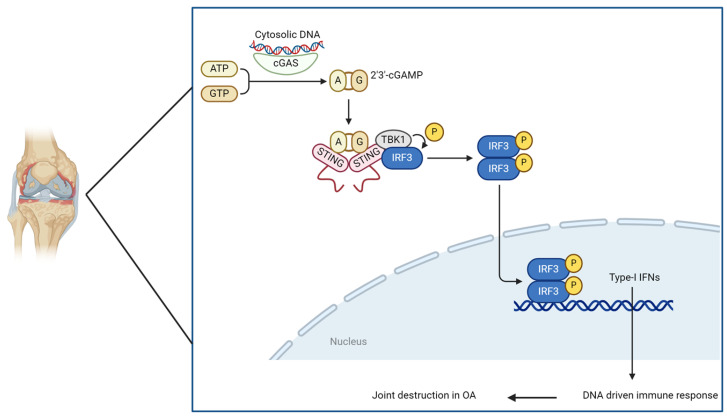
Role of cGAS-STING pathway in OA joint.

**Figure 5 ijms-25-07411-f005:**
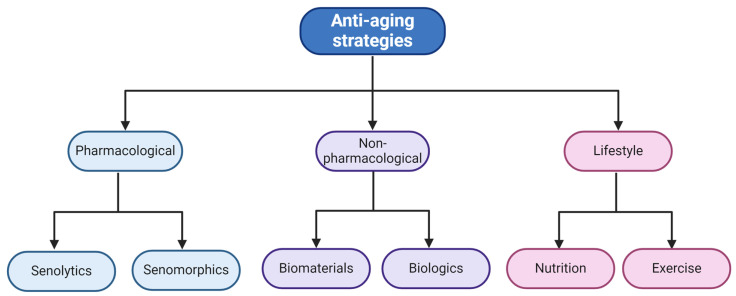
Regeneration interventions and anti-aging strategies for bone senescence.

**Table 1 ijms-25-07411-t001:** List of studies aimed at bone senescence.

Study Summary	NGS Methods	Key Findings	Key Genes with Elevated Expression	Reference
RNA profile expression in eight experimental models of cellular senescence commonly studied. WI-38 and IMR-90 fibroblasts, HUVECs and HAECs were cultured and utilized for downstream sequencing.	RNA seq, Illumina Hiseq 2500 with 2 × 150-bp strategy	50 elevated and 18 reduced transcripts and the identification of subsets of transcripts (both coding and non-coding) display shared expression patterns across a range of senescent cell models.	*SRPX*, *PURPL* (p53 regulator)	[111]
RNA expression profiling of fibroblasts and their senescence induced by 5-aza identified 3 epigenetically silenced pathways.	Illumina HiSeq 2000; strategy not specified	5-aza-induced senescence has been closely linked to alterations in the interferon/innate immunity pathway’s gene expression, and during immortalization, important regulators of this pathway are muted.	*IL-1α* and *IL-1β*	[112]
Use of multiple whole-transcriptome datasets, created by the authors or made publicly available, to characterise the heterogeneity of the programmed senescence.	Illumina HiSeq 2000 with 2× 150 bp	Demonstrates that ollowing senescence induction, the senescent phenotype identified for 55 genes at the core of the senescence-associated transcriptome is dynamic, changing at different intervals.	Upregulation of genes associated with G1 DNA damage checkpoint (*PLK3* and *CCND1*) and upregulation of *BCL2L2* (negative regulator of apoptosis)	[113]
Comprehensive analysis of the transcriptome and senolytic responses in a panel of 13 cancer cell lines rendered senescent by two distinct compounds.	HiSeq 2500; single-end 65 bp	Cell lines that were made senescent by two different substances showed that the senescence trigger has less of an impact on the composition of the SASP. The SENCAN gene expression classifier to detect senescence using machine learning was developed.	*IL6* and *CXCL8*	[114]
Gene set generation (SenMayo) consisting of 125 previously identified senescence/SASP-associated factors.	HiSeq 2000; strategy not specified	Provided a unique gene set (SenMayo) that can be utilized in bulk and scRNA-seq investigations to detect cells expressing high amounts of senescence/SASP genes. SenMayo rises with aging across tissues and species and is responsive to senescent cell clearance.	*CDKN1A/P21Cip1* and several SASP markers such as CCL2 and IL6 showed consistent upregulation with aging	[102]

## Data Availability

Not applicable.

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
