# Peer review of "Cellular Senescence and Inflammaging in the Bone: Pathways, Genetics, Anti-Aging Strategies and Interventions"

_ijms, 2024, doi:10.3390/ijms25137411_

Round 1

Reviewer 1 Report

Comments and Suggestions for Authors

I read this interesting and well written review on genes, pathways and mechanisms involved in bone senescence and inflammaging. I found it interesting and clear, but I suggest the authors to add an important section on bone senescence and inflammaging, clarifying also specific characteristics of the aging bone tissue including resident cells involved.  

Following a list of suggestions and comments:

Line 19, pag 1: typos for  'inflammaging'. Please correct it.

Line 71, page 2: Please specify what 'healthy life style’ generically refers to.

Line 88, page 2: Please add some examples for SAPS in bone (what specific proteins, cytokines and chemokines) 

Line 92, page 2: Please change 'can lead' to 'can cause'.

Line 117, page 3: Please ‘pathway’ instead of ‘patchway.’

Paragraph 3 ‘Pathways involved in senescence and inflammaging in bone’: In my opinion, the authors should specify which bone cells are involved in the senescence and inflammaging process: should we consider all the pathways described as being involved in all resident cells, or are there some specificities/differences? Readers less familiar with the topic should understand more easily which pathways are relevant for the different bone cells, and a brief explanation on it in this section may help the authors to improve their  understanding.

Line 267, page 8: Please add the correct number reference to the citation Ghosla et al (2021) as it misses. 

It would be interesting to know which cells are the senescent cells involved in the ARDs mentioned by the authors. Since this review focuses on these (specifically OA, OP and RA), the authors could spend a short statement on this crucial point, which could act as a glue to the multiple data provided. Indeed, it is a pity that the latter are not strongly linked to each other in this interesting review.

Finally, a brief explanation on bone aging and BM: what are the relevant features? This would contribute to make more organic the all review.

Reviewer 2 Report

Comments and Suggestions for Authors

Merin Lawrence et al. summarized some of the genes and signaling pathways associated with senescence in bone-related diseases. I have some suggestions to the authors:

l  Abstract: When you first mention BM (line 27), you should write the full name followed by abbreviation, such as bone marrow (BM). 

l  Some words need to be corrected on page 3, for example, Chromatin “remodeling” in Figure 1, should be “remodeling”. “pacthways” on line 117 should be “pathway”. “p35” on line 120 should be “p53”.  “ageing” on page 13 line 451 should be “aging”. 

l  Figure 2: The authors described p16, p21, and p53 pathways on page 4. However, the Figure 2 displays p16, p21, and p27, without p53. The author should list P16. P21, and p53 in the Figure 2.   Also on page 4 line 154, “In p53 dependent”, missed a word “pathway”, it should be “in p53 dependent pathway”. 

l  Figure 3: the author mentioned that mTORC1 inhibits autophagy by phosphorylating ULK1 on page 5 line 173. However, the figure 3 did not show ULK1. They should include ULK1 in the figure 3. 

l  When you mention a person’s name et al, you should include reference in the paper. For example, on page 5, line 177 (Zhang et al.), on page 5 line 179 (Pan et al.), on page 8, line 267 (Ghosla et al.), on page 8, line 273 (Khosla et al.), on page 9, line 283 (Ghosla et al.), on page 10, line 322 (Zhu et al.), on page 11, line 387(Chung et al.),  on page 12, line 418 (He et al.), on page 12, line 428 (Zhou et al.), on page 13, line 479 (A recent study), on page 13, line 483 (Fielding et al.), and on page 14, line 490 (Martel et al.) These studies need to be cited. 

l  Figure 4: The authors mentioned that STING pathway via downstream targets, including TBK1, IRF-3, and IKK on page 6. However, the figure 4 only display IRF3 and NF-kB, without TBK1. The author should include TBK1 in the figure 4. 

l  When you first mention a gene name, you should write the full name, then abbreviation. The DDR, OPG, TIMP1, MIG1, PRDX, IBGBP3 on page 7 need to include the full names of the genes. 

l  Table 1: the word “utilised” should be corrected as “utilized” on page 8 and page 9. 

l  The author should generate a Table 2 to summarize the pharmacological, non-pharmacological, and lifestyle strategies for combating senescence in bone. This include more detailed information of drugs, results, references associated with the studies. 

Comments on the Quality of English Language

Some of the words need to be corrected. 

Reviewer 3 Report

Comments and Suggestions for Authors

Dear Merin Lawrence and co-authors,

Thank you for the well prepared manuscript entitled "Cellular senescence and inflammaging in the bone: Pathways, 2 genetics, anti-aging strategies and interventions". Please find below my comments and questions of understanding:

Major comment:

·         The content of the manuscript provides an overview of senescence. However, there is little specific information about bone in this manuscript. It only mentions bone-related papers from time to time, but does not go into details The title made me expect much more. It is a nice overview over senescence, but there is no strong focus on bone. If there is something written over the bone or bone treatment, then it is not explained how the senescence is influenced by it. So one part of the whole story is always missing.

·         The signal paths (Chapter 3) are described briefly and concisely but sufficiently. However, their significance in relation to bone is only described in the case of OA. Furthermore, osteoarthritis is not a bone disease but a joint disease. There are huge differences between bones and joints, not only in their structure but also in the underlying biology. Is there no further data on osteoporosis, bone remodeling in general in old vs. young people or fractures? If not, the paper maybe should be focused on "Cellular senescence and inflammaging in osteoarthritis"

·         Page 10, line 319/320: “This is the class of therapeutics that selectively eliminate senescent cells to prevent or 319 alleviate several age-related conditions such as senile OP”. At the moment there are no senolytics which are used to prevent OP. Maybe this could be a possible treatment, but it’s not on the market. Do you know publications who deal with treatment of OP with senolytics? If yes, please add them. Otherwise this sentence is not correct.

Minor comments:

·         Page 1, line 27: What does the abbreviation “BM” mean? Bone marrow?  The explanation is missing in the abstract.

·         Page 1, line 33: One or more words are missing in the sentence “It has been closely linked with several conditions, age-related diseases (ARDs) and are established factors in reduced quality of life (QOL) in the elderly”

·         Figure 1; Bullet points below the word “Senescence Associated Secretory Phenotype”:  It would certainly be helpful to color the bullet points in the color in which the structures mentioned (e.g. lysosomes, chromatin) are shown in the drawing. That way you know what the "circles" in the drawing are supposed to be.

·         How does the biologics (page 13, Chapter 5.2.2) influences the senescence markers?

·         Does the lifestyle approaches in chapter 5.3 (page 13) have an influence on the bone quality?

Comments on the Quality of English Language

The quality of the English language is good. In one case, however, a sentence is not complete (see comment above).

Round 2

Reviewer 1 Report

Comments and Suggestions for Authors

The revised version is satisfactory. I have no further comment since the authors did a great job!